# Automated Computer-Assisted Medical Decision-Making System Based on Morphological Shape and Skin Thickness Analysis for Asymmetry Detection in Mammographic Images

**DOI:** 10.3390/diagnostics13223440

**Published:** 2023-11-14

**Authors:** Rafael Bayareh-Mancilla, Luis Alberto Medina-Ramos, Alfonso Toriz-Vázquez, Yazmín Mariela Hernández-Rodríguez, Oscar Eduardo Cigarroa-Mayorga

**Affiliations:** 1Department Advanced Technologies, UPIITA-Instituto Politécnico Nacional, Av. IPN No. 2580, Mexico City C.P. 07340, Mexico; ymhernandez@ipn.mx; 2CICATA-National Polytechnic Institute, Legaria 694, Mexico City C.P. 11500, Mexico; lmedinar1503@alumno.ipn.mx; 3Academic Unit, Institute of Applied Mathematics and Systems Research of the State of Yucatan, National Autonomous University of Mexico, Merida C.P. 97302, Yucatan, Mexico; atoriz98@comunidad.unam.mx

**Keywords:** asymmetry analysis, breast cancer, computer-aided diagnosis, early detection, mammography

## Abstract

Breast cancer is a significant health concern for women, emphasizing the need for early detection. This research focuses on developing a computer system for asymmetry detection in mammographic images, employing two critical approaches: Dynamic Time Warping (DTW) for shape analysis and the Growing Seed Region (GSR) method for breast skin segmentation. The methodology involves processing mammograms in DICOM format. In the morphological study, a centroid-based mask is computed using extracted images from DICOM files. Distances between the centroid and the breast perimeter are then calculated to assess similarity through Dynamic Time Warping analysis. For skin thickness asymmetry identification, a seed is initially set on skin pixels and expanded based on intensity and depth similarities. The DTW analysis achieves an accuracy of 83%, correctly identifying 23 possible asymmetry cases out of 20 ground truth cases. The GRS method is validated using Average Symmetric Surface Distance and Relative Volumetric metrics, yielding similarities of 90.47% and 66.66%, respectively, for asymmetry cases compared to 182 ground truth segmented images, successfully identifying 35 patients with potential skin asymmetry. Additionally, a Graphical User Interface is designed to facilitate the insertion of DICOM files and provide visual representations of asymmetrical findings for validation and accessibility by physicians.

## 1. Introduction

Breast cancer is a multifaceted disease with diverse clinical manifestations and prognoses. Among the common presentations are Invasive Ductal Carcinoma, Invasive Lobular Carcinoma, and Inflammatory Breast Cancer. These specific types of cancer are often associated with asymmetry between the right and left breasts, serving as early indications of the disease. The early detection of breast cancer is crucial to ensure successful treatment and improved survival rates. Breast cancer is the most common cancer among women worldwide, accounting for over 2 million new cases and 600,000 deaths annually [1]. The incidence of breast cancer varies widely across regions, with higher rates observed in developed countries. In Mexico, breast cancer is the leading cause of cancer death in women, with an estimated 28,000 new cases and 6000 deaths each year [2]. In recent years, efforts have been made to improve breast cancer diagnosis by accessing the population screening services in Mexico. The establishment of national breast cancer screening programs and the introduction of new technologies, such as digital mammography and breast ultrasound, have contributed to improved detection and treatment outcomes [3]. Thus, the early detection of breast cancer is crucial to ensuring successful treatment and improved survival rates. The early detection of breast cancer is crucial to improving survival rates and reducing the burden of the disease. Breast cancer screening programs, including mammography and clinical breast examination, are effective in reducing breast cancer mortality. However, access to screening and treatment remains a challenge in many low- and middle-income countries, where disparities in access to healthcare and cancer services exist [2,4].

While histopathology images provide valuable insights into the cellular composition of breast tissue, obtaining such data can be challenging in certain regions, limiting the widespread application of AI-based techniques [5]. To address this limitation, our research focuses on developing a computer system solely based on computer vision and morphological features for asymmetry detection in mammographic images. By leveraging mammogram images, which are widely available and commonly used for breast cancer screening, our approach aims to support advances in AI for breast cancer detection. In this context, we explore mammogram images over histopathology images in terms of availability and practicality. While histopathology images offer detailed cellular information, they may not always be easily accessible, especially as per current medical guidelines, histopathology is not typically recommended for cases classified via the Breast Imaging Reporting and Data System (BIRADS category 1 or 2) [6]. Also, In BIRADS 1 and 2 cases, asymmetry is usually considered benign and is not a cause for immediate concern. However, it is still important for radiologists to identify and document any asymmetry observed during mammogram evaluation. Regular follow-up mammograms may be recommended to monitor the stability of these findings over time [7,8]. On the other hand, mammogram images provide a non-invasive and widely accessible imaging modality for breast cancer screening, making them a more practical choice for developing AI-based systems.

While discussing the use of mammogram images, we also acknowledge the significance of the AI-based histopathology techniques proposed in the literature. These techniques have demonstrated valuable insights into the cellular and molecular characteristics of breast cancer. However, our study focuses on harnessing the potential of computer vision and feature extraction to develop an innovative approach that complements existing methods and supports early breast cancer detection.

Currently, medicine has several auxiliary systems in terms of diagnosis and treatment. Computer-Aided Diagnosis (CAD) is an advanced tool in the field of medical diagnosis, with rapid acceptance among the medical community. The primary objective is to support evidence-based medical diagnoses by leveraging statistical analysis and data processing techniques [9]. Regarding early breast cancer detection, CAD systems have emerged as relevant systems in the detection and analysis of mammographic images. The development of a CAD system specifically designed for the identification of asymmetries in mammograms based on morphology and skin thickness is a significant contribution to this field. The integration of CAD technology in the early diagnosis of breast cancer has the potential to enhance detection rates and reduce false positives. It serves as a complementary tool that aids in the identification of suspicious areas, allowing for timely intervention and treatment planning [10]. Figure 1 depicts a schematic representation showcasing the integration of a CAD system with mammography equipment. This seamless connection enables the real-time diagnosis of potential breast cancer cases at an early stage.

Once the result is obtained, it is displayed in a PDF report so that only the physician will have that information. Asymmetry holds potential biological significance and can be influenced by various factors, including genetics, hormonal fluctuations, developmental abnormalities, and lifestyle choices, without necessarily indicating the presence of tumors. For instance, during puberty, hormonal changes can lead to the uneven growth of breast tissue, resulting in variations in breast size and shape. Moreover, genetic factors can also contribute to variations in breast size and shape, as some individuals may inherit larger or smaller breasts from their parents. Overall, breast asymmetry is a natural occurrence influenced by a range of biological and lifestyle factors. A study conducted by Kayar et al. showed a correlation between morphological asymmetry and breast cancer. Statistically significant evidence was observed linking breast asymmetry and cancer when the breast asymmetry ratio exceeded 20%. This association held true in all age groups, specifically in the 40–69 age range, with a 99% confidence interval [11]. Although generally considered normal, significant asymmetry can cause physical and emotional discomfort, and more importantly, serve as an indicator of potential tumor development [11,12].

One of the key factors in the early detection of breast cancer is identifying asymmetry between the left and right breasts. Asymmetry can be a potential sign of breast cancer, as it may indicate changes in breast tissue composition and structure. Therefore, asymmetry analysis plays a critical role in breast cancer diagnosis and treatment [12,13]. Asymmetry is typically observed in one of the standard mammographic projections and is not visible in additional views. This finding commonly represents the overlap of fibroglandular tissue and summation artifacts, occurring in approximately 80% of cases. It often contains interspersed fat within the affected area. In recent years, several approaches have explored methods for asymmetry analysis, including shape analysis and skin thickness measurement. Breast asymmetry is an effective tool for breast cancer diagnosis and classification [14]. Centroid-based shape analysis involves the extraction of features related to breast shape from mammograms, which can then be used to train machine-learning models for breast cancer classification. In addition to shape analysis, skin thickness measurement has also been identified as an important factor in breast cancer asymmetry classification. Studies have shown that skin thickness measurements can provide valuable information about breast tissue composition and structure, which can help to identify potential areas of concern [15]. Table 1 presents a summary of state-of-the-art breast asymmetry studies.

Based on the information provided in Table 1, several approaches can be observed for assessing asymmetry based on certain contralateral features. However, the extraction of these features may rely on human criteria, which could lead to false positives due to subjectivity, overlapping features, human fatigue, or diagnostic challenges. On the other hand, Artificial Intelligence (IA) techniques such as Deep Learning may overcome these issues; however, the selection of feature extraction from digital mammograms plays a crucial role in training prediction models. Therefore, in this paper, we propose an approach for feature extraction related to breast shape based on Dynamic Time Warping (DTW) and skin thickness based on Growing Region Seed (GRS).

The computed features from our analysis focused on morphological analysis and breast skin segmentation, and serve as essential building blocks that could enhance current AI models. By leveraging advanced deep learning frameworks and multi-scale feature fusion approaches, we envision these features contributing to a more robust analysis of segmentation, tumor detection, and classification. Additionally, our findings hold promise for applications like AI-based classification using histopathological and textural images, as a perspective for this paper. These advancements demonstrate the potential of AI models to enhance accuracy and efficiency in diagnosing and classifying medical conditions, which is crucial for clinical decision-making and patient outcomes. These features can be integrated into a machine learning pipeline, such as in the approaches seen in the SegR-Net, RAAGR2-Net, and Deep Learning classification studies. By incorporating the identified features into machine learning models, we envision an improvement in the automated classification of mammograms, allowing for the accurate and efficient detection of potential breast asymmetry [21,22,23,24]. Additionally, our research lays the groundwork for exploring similar AI-based approaches in other medical-related research problems, such as retinal vessel segmentation and brain tumor classification. As AI continues to be a hot topic in medical research, our study contributes to the growing body of knowledge on the potential applications of AI in medical image analysis. The exploration of morphological shape and skin thickness analysis as valuable features for breast cancer detection opens avenues for further research and collaboration between medical professionals and technology experts. Our findings serve as a steppingstone toward developing more sophisticated AI models that can aid in early detection and improve treatment outcomes for breast cancer patients and potentially other medical conditions.

We aimed to research the potential use of morphological shape and skin thickness analysis in mammographic images for the early detection of breast cancer. Our hypothesis aims to explore the potential of a comparative analysis between the centroid–perimeter radius and breast skin segmentation as a reliable method for detecting potential breast asymmetry. We have utilized Dynamic Time Warping (DTW) for shape analysis and the Growing Seed Region (GRS) method for breast skin segmentation to assess their accuracy. Despite most mammograms not being classified with asymmetry by radiologists, the differences obtained with DTW and skin thickness analysis could indicate possible asymmetry patterns. It is essential to emphasize that our results are solely descriptive features that could be utilized in future works, particularly in the development of learning models for early breast cancer detection. The results of our investigation could have implications for the early diagnosis and monitoring of breast cancer, ultimately leading to improved patient outcomes. However, it is essential to consider the limitations of our study, such as the absence of a classification learning model and the reliance on radiologist-identified cases for validation. We discuss these limitations in more detail to provide a comprehensive understanding of the scope and potential future directions of our research.

The paper’s structure is as follows: Section 2 describes the patient dataset, image preprocessing, breast structural and thickness analysis, and the Graphical User Interface (GUI) for the CAD system design. Section 3 of the paper presents the results and their corresponding analyses, while Section 4 and Section 5 are dedicated to the discussion and conclusion, respectively.

## 2. Materials and Methods

This section provides a detailed explanation of the procedure employed to identify potential cases of asymmetry based on the analysis of breast morphology and skin thickness differences between contralateral breasts. The procedure involves several steps, including data collection, image preprocessing, feature extraction, and statistical analysis. The methodology ensures a systematic and rigorous approach to identifying and classifying asymmetry in breast images, contributing to the early detection of potential abnormalities. Figure 2 depicts a flowchart for this section.

### 2.1. Patient Data-Set

The clinical dataset includes the Medial Lateral Oblique (MLO) and Craniocaudal (CC) projections, which represent the breast shape along the sagittal and coronal planes, respectively. Additionally, the dataset includes views from the axial/transverse plane for both left and right breasts. The MLO projection provides an oblique view of the breast from the medial to lateral direction, while the CC projection offers a view from the cranial to the caudal direction. The dataset consists of 114 patients in DICOM format from 30 to 71 years old, diagnosed with BI-RADS from B1 to B6, with an average age of 55 years old. Table 2 presents the distribution of patients regarding the BI-RADS category. The data used in this study were sourced from the reports obtained from the University of Pittsburgh Text Information Extraction System (TIES) [25,26]. TIES is a database with anonymized information that includes around 24 million radiology reports generated between 2003 and 2015 at all 18 hospitals affiliated with the University of Pittsburgh Medical Center (UPMC).

Spatial transformation through normalization aligns mammographic images to a standardized resolution, ensuring the consistent and accurate analysis of spatial descriptive features. This process enhances the reliability of comparisons between images, facilitating the detection of potential breast asymmetry patterns and contributing to early breast cancer detection.

### 2.2. Image Pre-Processing

Digital mammography is an imaging technique used for the early detection and diagnosis of breast cancer, replacing traditional film-based mammography by capturing digital images of the breast using specialized X-ray equipment. In digital mammography, the acquired images are immediately available for review on a computer screen, providing higher image quality, and enabling radiologists to visualize subtle abnormalities more clearly. However, digital images can be enhanced, manipulated, and magnified to assist in accurate interpretation and analysis. Commonly, digital mammography has a .dcm format extension, which stands for Digital Imaging and Communications in Medicine (DICOM), which is a standard format for medical images that enables the exchange of images and patient information.

DICOM images can be processed using image processing techniques for various applications such as diagnosis, treatment planning, and research. Nevertheless, due to variations in equipment standards, the spatial resolution of mammographic images may pose challenges in accurately describing statistical features related to spatial characteristics. To mitigate this issue, a normalization process was implemented to ensure consistency across all images obtained. The resolution of the images was standardized to 521 × 512 pixels, considering the variation in spatial resolution among the mammograms, as there was no uniform standard for acquisition. However, upon careful examination, it was deemed more appropriate to align the images to a resolution of 512 × 512 pixels to ensure consistent analysis of spatial descriptive features. Consequently, the normalization process was adjusted accordingly, aligning all mammograms to the common resolution of 512 × 512 pixels. This standardization facilitates reliable and meaningful comparisons by ensuring consistent spatial dimensions for the accurate analysis of spatial descriptive features. Spatial transformation through normalization aligns mammographic images to a standardized resolution, ensuring consistent and accurate analysis of spatial descriptive features. This process enhances the reliability of comparisons between images, facilitating the detection of potential breast asymmetry patterns and contributing to early breast cancer detection.

A second issue concerns annotations made by radiologists, physicians, or the imaging equipment itself, which include patient information, projections, and metadata. These interferences can disrupt feature extraction algorithms and bias prediction thresholds. To address this, a threshold value (T = 100) was calibrated on the grayscale level scale, ensuring the identification of all relevant regions above the threshold. Subsequently, the largest connected component, which consistently corresponds to the breast, was segmented. This approach aimed to mitigate the influence of annotation variability and establish a standardized basis for subsequent analyses. To identify the breast region, we utilized the concept of the largest connected component, which refers to the region with the highest number of connected pixels in the image. As annotations are typically small, isolated islands, the largest connected component consistently represented the breast region with the most extensive pixel count. By employing this approach, we ensured that the breast area was accurately segmented, mitigating the influence of annotation variability and establishing a standardized basis for subsequent analyses.

### 2.3. Breast Structural Analysis Based on Dynamic Time Warping

Breast asymmetry structure serves as a significant indicator that can potentially alert medical professionals to the presence of early-stage cancer. While it is natural for some degree of asymmetry to exist between the breasts of most individuals, a significant deviation from the expected level of asymmetry can raise concerns. In the context of cancer detection, an abnormal level of asymmetry can signify the presence of underlying malignancies [11].

The approach of DTW in our study is well-established and widely used in various fields, including speech recognition, image processing, and pattern recognition. DTW allows for finding the optimal alignment path between two sequences, considering temporal or spatial distortions between them. The algorithm follows a right, diagonal, and down approach related to three fundamental conditions: boundary, monotonicity, and continuity [27]. The boundary condition ensures that the warping path must begin with the starting points of both signals and end with their endpoints. The monotonicity condition preserves the relationship in time, meaning a point from signal A in position x_n must not be aligned with a point from signal B in position x_n − 1. This constraint is the main reason why the approach follows a right, diagonal, and down path. Lastly, the continuity condition limits the number of steps a point from signal A may take to align itself with a point in signal B. While other alignment methods like left, diagonal, and up approaches are theoretically possible, they are less commonly used in image analysis and sequence comparison. In our application of breast image asymmetry analysis, the right, diagonal, and down approach offers better alignment accuracy. We have conducted a thorough review of the related literature and found numerous studies that utilized DTW with the right, diagonal, and down approach for similar applications, such as feature extraction and sequence alignment in medical imaging and time-series analysis [27,28]. Introducing the left, diagonal, and up approach might lead to less accurate alignment and could compromise the reliability of our asymmetry analysis results. Therefore, in alignment with the existing literature and to ensure robust and accurate results, we maintain the use of the right, diagonal, and down approach in our study of breast image asymmetry analysis.

An approach to analyzing breast asymmetry is to compute a collection of radii formed between the centroid and the skin perimeter. Once the centroid is computed, the radius is retrieved based on the angle as the argument between [0, π] radians in steps of one radian, creating a distance for each orientation between the centroid and perimeter of the skin. This supports uniformity and ensures that the image is created using the outermost breast pixels. To handle variations in breast size and shape among different DICOM images, the distances between the centroid and the skin perimeter were normalized to fall within the range of [0, π]. This normalization process ensured that the spatial features used for evaluation were consistent and comparable across all images, regardless of their individual radii. By generalizing this approach to the entire dataset, we were able to effectively evaluate the morphological asymmetry in breast skin thickness and its potential significance for early breast cancer detection.

Nevertheless, an important factor to consider is the orientation of the breast. Hence, both images (i.e., left, and right breast images) were adjusted to the same orientation, i.e., towards the right. The hypothesis proposes that the set of distances would differ between left and right views of the breast. To maintain consistency and ensure direct comparability, all images were oriented toward the right. Although orienting the image set towards the left is feasible, it would necessitate image reprocessing and algorithm adjustments to adjust the left-oriented images. Our specific focus on the right-oriented images allowed us to investigate the impact of asymmetry on the right breast and evaluate the effectiveness of our proposed approach. Figure 3 presents the flowchart used to obtain a collection of distances between the centroid and the edge of the breast, depicted in Figure 4. Appendix A provides the pseudocode for the algorithm.

After obtaining the set of distances between the centroid and the edge of the breast at the angles ranging from 0 to π radians, a vector comprising these distances is extracted. This vector serves the purpose of enabling a structural comparison of shape between the contralateral breasts. It is assumed that the presence of asymmetry is indicated when shape similarity between the two vectors is lacking. To assess the similarity between both vectors, Dynamic Time Warping (DTW) can be used to assess alikeness.

DTW is a technique used to measure the similarity between two time series with varying lengths. However, in the context of breast asymmetry detection, the vector of distances can be used to represent the shape characteristics of the breasts [29].

Breasts can exhibit variations in size while maintaining similar underlying structural characteristics. When analyzing the vector of distances, the objective is to identify corresponding points on the curves of the left and right breasts. These corresponding points represent areas where the radii’s values align or coincide, indicating similarity in shape or structure. To achieve this, DTW allows a warping path between the curves. In the context of breast asymmetry detection, the angle values serve as the basis for constructing the time series. By employing DTW, the algorithm searches for the most similar alignment between the curves of two shapes, i.e., the left and right breasts. It identifies points along the curves that exhibit similar shapes and structures, regardless of their temporal or spatial position. These points of coincidence or matching correspondence indicate areas where the breasts demonstrate asymmetry. By analyzing the extent of similarity or dissimilarity in the correspondences, it becomes possible to quantify the degree of asymmetry between the breasts. The greater the deviation in the alignment or the lack of matching correspondences, the higher the likelihood of significant asymmetry. Therefore, through the application of DTW to the vector of distances, the algorithm aims to identify and characterize asymmetry by seeking points of coincidence that align the curves of the left and right breasts, even in the presence of size differences.

The DTW algorithm involves several steps. First, a cost matrix is computed to measure the dissimilarity between each pair of elements in the sequences. The cost between two elements is typically determined using a distance metric, such as the absolute difference or squared difference. Let d = [d_1_, d_2_, …, d_n_] and a = [a_1_, a_2_, …, a_m_] be two input sequences of distances and angles, respectively, of lengths n and m. The cost matrix C of size (n + 1) × (m + 1) is defined. Then, it is initialized to C[0][0] = 0, and the cost between x[i] and y[j] is computed for each C[i][j]. The cost between both elements x[i] and y[j] is typically defined as the absolute difference or squared difference.

Next, dynamic programming is applied to find the optimal alignment. The cumulative cost matrix is generated by considering three possible moves: moving horizontally, diagonally, or vertically. The cumulative cost at each position in the matrix is updated based on the cost of the current elements and the minimum of the adjacent cumulative costs. The cumulative cost starting from C[1][1] considers three possible moves: right, diagonal, and down. For each element C[i][j], the minimum cost is computed and C[i][j] is updated as the cost between x[i] and y[j] plus the minimum of the three adjacent elements.

Once the cumulative cost matrix is computed, the optimal path is determined by backtracking from the bottom-right corner to the top-left corner. This path represents the alignment that minimizes the total cost and provides information about the similarity or dissimilarity between both breasts. This alignment provides insights into the similarity or dissimilarity between two sequences and is useful for pattern recognition, which is the goal of this paper.

### 2.4. Breast Thickness Analysis Based on Growing Region Seed

Breast skin thickness has been found to correlate with breast asymmetry and cancer. Studies based on ultrasound imaging were found to be significantly different in patients with breast cancer compared to those without cancer and have shown increased thickness of the breast skin [16].

The process of skin segmentation involves setting a threshold for the breast since skin typically has a higher intensity than the internal tissues of the breast. While a few regions of the perimeter may stand out, they may not fully define the skin. However, these regions are used as “seeds” to expand the Region of Interest (RoI). To standardize the gray intensity of each mammogram, a normalization process was conducted, considering the number of bits present in the image, which was determined from its maximum intensity value. 

The first step was to obtain a mask by the Otsu method to eliminate low-contrast tissue. Then, erosion was performed on the binary image using a kernel with size 8, which determines the border thickness. By comparing the mask concerning the border coordinates, we obtain the pixels that changed after erosion, thus retaining only the boundary of the breast. Then, a window with zero matrices with dimensions (n + 2) × (m + 2) was created, and a line of pixels at the top, bottom, left, and right, to evaluate the border pixels during region growth without requiring special functions. The output of these steps is an image with the same size as the original mammography but with these black border pixels, allowing post-processing instead of the original image without encountering border-related issues. To mitigate the issue of jagged edges observed in the mask when directly applying the Volume of Interest Look-Up Table (VOI LUT) technique, an alternative approach was adopted. However, the resulting mask was subsequently applied to the mammograms that had undergone normalization using the VOI LUT method. The VOI LUT is a technique employed in medical imaging to enhance the visualization of image data by mapping pixel values to a desired display intensity range. It is an important component in the post-processing pipeline for adjusting the windowing and leveling of medical images. The VOI LUT operates by applying a linear transformation to the pixel values, thereby adjusting their brightness and contrast characteristics. This allows medical professionals to better perceive subtle details and abnormalities in the image. The application of the VOI LUT is typically performed by specific standards and algorithms, ensuring consistent and reproducible results across different imaging systems. By effectively optimizing the display of medical images, the VOI LUT contributes to the accurate interpretation and diagnosis of various medical conditions [30].

Afterward, the coordinates where the seeds are located are obtained to proceed with the expansion function. At this point, we evaluate whether the pixel should belong to the growing region, or whether this pixel was already taken as part of the mask. Then, this pixel is stored in the mask sub-image, and a counter starts to mark a value of 1 for each pixel evaluated, preventing redundant evaluations, and significantly speeding up processing. At this point, a list of the seed coordinates and their neighboring pixels is created. Four conditions are evaluated for the neighboring pixels to determine if they should be part of the RoI:The pixel must not be black.Its difference from the seed pixel must be below 200, set as the high threshold.Its difference from the seed pixel must be above 1, set as the low threshold.Its Euclidean distance from the first seed in the cycle must not exceed 4 pixels deep.

If all these conditions are met, and the pixel has not been discarded before, it is attempted to add the pixel to the mask. Then, the current seed is removed, and the list is evaluated to find new seeds that may have appeared until no further derivations remain with all other seeds along the border. At the end of this loop, the border mask is retrieved. Figure 5 depicts the pseudocode for region growth by seed expansion. Appendix B provides the pseudocode for the algorithm.

In summary, the idea behind the algorithm is to use skin thickness as a preliminary indicator for detecting edges in an image. Skin thickness can be defined as the degree of brightness or darkness of the pixels in the skin region. By identifying the skin pixels in an image, the algorithm can then expand the mask to include neighboring pixels with similar intensity values. This assumes that skin pixels should have similar intensities due to the uniformity of skin tissue density. Several methods have been proposed for skin breast segmentation; however, unsupervised, or semi-supervised, segmentation processes must be validated against a set of ground truth images. This process was validated with the Average Symmetric Surface Distance (ASSD) method and a second validation was performed with the Relative Volumetric Distance (RVD) method. To assess the performance of the GRS algorithm for segmentation, ASSD was chosen as the first validation metric due to its ability to measure the average distance between corresponding points on segmented surfaces. By comparing the segmented results to ground truth data, ASSD provides valuable insights into the accuracy of surface delineation. However, since ASSD focuses solely on surface distances, it may not fully capture volumetric differences. Therefore, as a complementary measure, we employed RVD as the second validation metric. RVD evaluates the volumetric differences between segmented regions, considering both surface and internal variations. By using both ASSD and RVD, we gain a comprehensive understanding of how well our algorithm performs in terms of both surface delineation and volumetric consistency. This two-fold validation approach ensures a more robust evaluation of the growing region seed algorithm’s effectiveness in accurately segmenting regions of interest.

ASSD is a metric used to evaluate the similarity between two shapes, typically in the field of medical imaging or computer vision [31]. It quantifies the average distance between corresponding points on segmented skin. Let S1 be the skin image segmented by GRS and *S*2 represent the radiologist segmentation, these being the two shapes that need to be compared. Then, the corresponding point should be identified. Let *P(S*1*)* denote a set of surface points of *S*1, so *P*1 *=* {*p*_1_, *p*_2_, *…*, *p_n_*} is the set of corresponding points on *S*1, and *P*2 *=* {*q*_1_, *q*_2_, *…*, *q_n_*} is the set of corresponding points on *S*2. The Euclidian distance $d(p_{i},q_{i})$ between each pair of corresponding points is computed for calculating the sum of distances. The shortest distance of an arbitrary pixel p to *P*(*S*1) is defined as Equation (1).
(1)$$d(p,\;P(S1))={min}_{P1\in P(S1)}\;p-P1$$

The ASSD is computed by dividing the sum of distances by the total number of corresponding points. The resulting value of ASSD represents the average distance between corresponding points on the two surfaces, indicating the overall similarity or dissimilarity between them, presented in Equation (2). A smaller ASSD value indicates a closer match and higher similarity, while a larger value suggests greater dissimilarity between the surfaces.
(2)$$ASD(S1,S2)=\frac{1}{\left|P(S1)\right|+\left|P(S2)\right|}\left(\sum_{P1\in P(S1)}d(P1,P2)+\sum_{P1\in P(S1)}d(P2,P1)\right)$$

On the other hand, (RVD) is a metric used to assess the similarity or dissimilarity between two segmented RoIs in medical imaging [32]. It provides a quantitative measure of the volumetric differences between the two regions. Let *V*1 be the segmented skin image and *V*2 represent the radiologist segmentation. First, the volume difference (*VD*) between *V*1 and *V*2 is calculated by taking the absolute difference: *VD = |V*1 *− V*2*|*. This represents the absolute volume difference between the two regions. Next, the average volume (*AV*) of the two regions is computed as *AV* = (*V*1 *+ V*2)/2. The AV serves as a reference value for normalization. Finally, the RVD is obtained by dividing the volume difference (*VD*) by the average volume (*AV*) in Equation (3). Also, let *A* and *B* be two classes.
(3)$$RVD=\frac{\left|B\right|-\left|A\right|}{\left|A\right|}=\frac{VD}{AV}$$

The RVD value represents the proportion of the volume difference relative to the average volume. A smaller RVD value indicates a higher similarity or overlap between the volumes, while a larger value suggests greater dissimilarity. The RVD method provides researchers with a quantitative measure to assess the volumetric differences between segmented regions, enabling the objective evaluation of segmentation algorithms and the comparison of segmentation results.

### 2.5. Graphical User Interface for CAD System in Medical Diagnosis

In this section, we describe the Graphical User Interface (GUI) design for a specialized Computer-Aided Diagnosis (CAD) system in the context of medical imaging. Specifically, the GUI is designed to facilitate the diagnosis of breast asymmetry using mammography images. This section focuses on the key features and functionalities of the GUI, emphasizing its role in enabling seamless interaction, data input, image loading, and report generation.

The GUI encompasses various essential components tailored to the specific requirements of medical diagnosis. It offers an intuitive and user-friendly interface through which medical professionals can input vital patient information, including name, surname, age, weight, and height. These data play a crucial role in maintaining accurate patient records and facilitating personalized analysis. A fundamental aspect of the GUI is its ability to load mammography images for both the left and right breast. This feature streamlines the process of importing relevant images into the CAD system, ensuring efficient analysis and evaluation. Through a well-designed file selection mechanism, medical professionals can easily access and integrate the necessary image data into the CAD system for further examination.

Once the mammography images are loaded, the CAD system performs a comprehensive analysis using advanced image processing techniques, feature extraction, and pattern recognition algorithms. The GUI presents the diagnostic results in a concise and visually informative manner, allowing medical professionals to interpret and evaluate the findings effectively. The aim is to provide accurate and reliable assessments of breast asymmetry, aiding in the subsequent diagnosis and treatment planning process. In addition to real-time analysis, the GUI enables the generation of detailed PDF reports summarizing the patient’s information, image analysis results, and diagnostic outcomes related to breast asymmetry. These reports serve as valuable resources for medical practitioners, providing a comprehensive overview of the patient’s condition and facilitating effective communication and collaboration among healthcare professionals. Figure 6 depicts the steps of the GUI system.

In conclusion, the GUI developed for the CAD system in this study offers an intuitive and user-friendly interface specifically designed for medical professionals involved in breast asymmetry diagnosis using mammography images. By streamlining data input, image loading, analysis, and report generation, the GUI enhances the overall functionality and usability of the CAD system. The integration of advanced image processing algorithms within the CAD system ensures accurate and reliable diagnostic outcomes. Future research in this domain should focus on expanding the system’s capabilities, exploring additional diagnostic parameters, and leveraging emerging technologies to further improve the accuracy and efficiency of breast asymmetry detection.

## 3. Results

### 3.1. Breast Shape Asimetry Analysis

The results point to several cases with asymmetry as inferred from the centroid–perimeter radius analysis, pointing to possible asymmetry for a DTW > 0.08 in an index from [0, 1] with a standard deviation of 0.19. From a total of 114 patients, 23 cases were found that could indicate asymmetry, and these are presented in Table 3.

Figure 7 shows mammography images of a patient with BI-RADS 4 exhibiting contralateral asymmetry. In this particular case, the result reveals the presence of noticeable asymmetry between both breast pairs, primarily within the first quadrant of the tomography. This observation implies that a substantial disparity should exist when comparing the radius projected from the centroid to the perimeter. To facilitate this analysis, it becomes essential to define a specific region where the centroid of a closed and solid geometry can be accurately calculated. In this regard, the previously established segmentation mask proves useful as it encompasses the relevant coordinates necessary for this calculation. 

To test the hypothesis regarding the disparities in centroid–perimeter distances, a comparison was conducted between the two geometries, as depicted in Figure 8. In Figure 8a, the centroid was determined to have coordinates (433,202), while in Figure 8b, the centroid coordinates were found to be (95,288). These coordinate values provide crucial information for further analysis and assessment of the observed differences in the context of the hypothesis.

Figure 9 shows the same BIRADS 4 representative cases of the presence of symmetry with a 0.24 × 10^−4^ DTW index; on the other hand, we can observe in Figure 10 the inequality between curves that reflect the geometry of the breast for the patient with BIRADS 2—even the retraction of the nipple is appreciable in the right breast, with a 0.22 DTW index. By comparing these parameters between both breasts, healthcare can identify differences in breast shape and size, and determine a degree of asymmetry.

According to the results of the database analysis, a DTW index less than 0.08 could indicate asymmetry due to shape or even factors such as nipple retraction. Figure 11 displays the results of the DTW analysis conducted on each patient, while Figure 7 shows validation results according to the Confusion matrix. The accuracy of the algorithm was 82%.

Figure 12 presents the results according to the DTW index for each of the 114 patients.

### 3.2. Breast Skin Segmentation Analysis

Breast skin segmentation enables a comprehensive evaluation of breast morphology, for the detection and characterization of asymmetrical regions. This information is crucial in the early detection and diagnosis of breast cancer, as asymmetry can be a significant indicator of the presence of tumors or other abnormalities [14]. Figure 13 presents the results of the GSR method.

To identify the optimal combination of upper and lower thresholds, erosion kernel size for seed generation, and growth distance, a parametric study was conducted using the data presented in Table 4. This study aimed to determine the parameter values that yield the most accurate and robust results in the segmentation process.

Table 5 presents the validation results of GRS-segmented images compared to those segmented by specialists. The validation metrics of ASSD and RVD were used for comparison to identify the most favorable combination that closely resembles the specialist-segmented images.

Once the optimal parameter combination was determined, the study focused on identifying cases of asymmetry based on the difference in skin thickness between contralateral breasts. Out of the 114 cases analyzed, the GRS algorithm detected 35 cases of asymmetry, among which 20 patients were previously labeled according to the radiologist’s criteria. At this juncture, it is crucial to highlight that certain cases of asymmetry attributed to variations in skin thickness may evade detection through mere visual examination by specialists. This underscores the significance of employing the algorithm, as it holds the potential to provide valuable alerts about the early diagnosis of breast cancer. The detection of these asymmetrical cases further validates the effectiveness of the GRS algorithm in assisting radiologists and facilitating the early detection of potential breast abnormalities. 

Figure 14a presents the case with the lowest asymmetry in skin thickness difference, with an average difference of 0.04 × 10^−2^, while Figure 14b shows the case with the highest asymmetry, with an average of 1.119.

Table 6 presents the performance of the GRS algorithm compared to reference images (182 images). The validation of the algorithm was conducted using the ASSD and RVD metrics. This analysis allows for a comprehensive assessment of the algorithm’s accuracy and effectiveness in quantifying the similarity between segmented images and reference data.

Figure 15 illustrates the level of asymmetry based on the average difference in thickness for each patient, where the line indicates the average difference that serves as the decision threshold.

This analysis offers valuable insights into the magnitude of asymmetry observed in the measurements of breast skin thickness. By setting a threshold based on the average skin thickness difference, it becomes feasible to classify patients based on their asymmetrical or symmetrical characteristics, thereby assisting in the detection of potential abnormalities. These findings may contribute to a better understanding of the variations in breast skin thickness and provide a useful tool for assessing breast health and identifying potential pathological conditions.

### 3.3. Graphical User Interface Design for Computer-Aided Diagnosis

This section presents key features of the GUI, including successfully capturing and organizing essential patient information, such as name, surname, age, weight, and height, ensuring accurate association with the corresponding medical images. Users can conveniently load mammography images for both the left and right breasts, enabling detailed examination and analysis. Advanced image processing techniques were incorporated into the GUI to preprocess the loaded images, effectively removing unwanted annotations, and enhancing the clarity of the breast structure (Figure 16).

A significant feature of the GUI was the generation of PDF reports, which encompassed the loaded images, patient data, and diagnostic results. This streamlined the reporting process, allowing medical professionals to access a comprehensive summary of the analysis performed by the CAD system. The reports provide a clear representation of the identified asymmetry, or lack thereof, aiding in the detection of potential breast abnormalities.

Overall, the evaluation of using the GUI in the CAD system in medical diagnosis shows its efficacy in enhancing the workflow of medical professionals. The intuitive design and integration of essential features, such as patient data input, image preprocessing, and report generation, contributed to an efficient and accurate diagnosis. The GUI’s successful implementation paves the way for improved medical decision-making and enhanced patient care in the field of breast diagnostics.

## 4. Discussion

To improve the comparison analysis for the centroid–perimeter radius and breast skin segmentation with the GRS method, several approaches can be employed. One possible method is the use of machine learning algorithms, which can learn from a large dataset of mammograms to identify potentially cancerous areas accurately. Another approach is to incorporate texture analysis techniques, such as the co-occurrence matrix, into the segmentation process to improve the accuracy of the segmentation. Moreover, using a combination of different segmentation techniques can also enhance the accuracy of the analysis.

To validate the results, ground-truth annotations are necessary, which can be obtained through biopsy or histopathology. Furthermore, the use of multiple radiologists to assess mammograms can increase confidence in the results. The integration of CAD systems can also improve the validation process by providing a second opinion on potential cancerous areas. In terms of medical correlation, the early diagnosis of breast cancer can significantly increase the chances of successful treatment. The accuracy of the comparative analysis of the centroid–perimeter radius and breast skin segmentation can aid in the detection of potential cancerous cases, which can then be confirmed through biopsy or histopathology.

Our study primarily focused on feature extraction and validation, rather than the implementation of a classification learning model. As a result, we did not have conventional control data. Instead, we applied the cases identified by radiologists as potential asymmetry, using them as reference points for our validation process. This approach allowed us to assess the accuracy and performance of the feature extraction methods utilized in our system. By employing the identified cases as validation data, we could evaluate the effectiveness of our proposed techniques in detecting possible breast asymmetry. It is crucial to underscore that the main goal of our study was to demonstrate the feasibility and reliability of the feature-based preliminary asymmetry detection system, paving the way for future advancements in developing more advanced classification models.

From a medical standpoint, the early detection of breast cancer is crucial for successful treatment outcomes. The accurate comparison analysis may serve as an auxiliary tool in identifying suspicious cancer cases that specialists may not assess due to human factors. However, it is important to note that the analysis presented in this study should be complemented by further clinical investigations and follow-up studies. Biopsy or histopathology remains the gold standard for definitive diagnosis, and the integration of our CAD system with these procedures can provide a comprehensive diagnostic approach. Overall, this research highlights the potential of both characteristics for early breast cancer detection and underscores the relevance of their integration into a Graphical User Interface designed for effective utilization in the healthcare field. The integration of machine learning algorithms, texture analysis techniques, and multiple segmentation methods may enhance analysis accuracy. Accurate and reliable analysis methods may enable an early diagnosis to have the potential to significantly improve treatment outcomes for breast cancer patients.

While our research has demonstrated the effectiveness of feature extraction in detecting possible breast asymmetry, there are certain limitations to consider. One of the key limitations is the lack of implementation of a classification learning model in our study. While we focused on feature extraction and validation, the absence of a classification model may limit the direct application of our findings for real-time diagnosis. Future work should aim to integrate machine learning algorithms into our system to enable the automated classification and accurate identification of potentially cancerous areas. Another limitation is the reliance on radiologist-identified cases as reference points for validation. While this approach served our purpose, it introduces a level of subjectivity that may affect the generalizability of our results. To address this limitation, future studies could incorporate ground-truth annotations obtained through biopsy or histopathology, providing a more objective basis for evaluation. Furthermore, the current study employed a specific set of segmentation techniques for breast skin and shape analysis. Alternative approaches, such as texture analysis techniques using co-occurrence matrices, could be explored to enhance the accuracy of segmentation and feature extraction. Lastly, our research employed a dataset with a limited number of patients, and may not fully represent the diversity and complexity of breast cancer cases. A larger and more diverse dataset would provide a more comprehensive evaluation of the proposed system’s performance and generalizability.

## 5. Conclusions

In conclusion, our research contributes to the advancement of early breast cancer detection through feature extraction and validation. The study underscores the significance of comparing contra-lateral breasts for asymmetrical analysis, focusing on structural shape and skin thickness. The findings emphasize the potential clinical applications of our proposed system and the need for further exploration in this field. Advancements in technology and machine learning offer the opportunity to enhance structural analysis for early detection and monitoring. While our study primarily focused on feature extraction and validation, future research should explore the integration of machine learning algorithms for automated diagnosis and improved accuracy.

Our study focused on developing a computer system for asymmetry detection in mammographic images, using Dynamic Time Warping (DTW) for shape analysis and the Growing Seed Region (GRS) method for breast skin segmentation. We observed a notable presence of asymmetry in 23 cases subjected to structural analysis, with a DTW index > 0.08 and an accuracy of 83%. The GRS-based skin segmentation achieved an average accuracy of 90.47% (ASSD) and 66.66% (RVD), successfully identifying 35 patients with potential skin asymmetry. However, we acknowledge certain limitations in our research. Notably, the absence of a classification learning model may limit the real-time application of our findings. Future work should aim to incorporate machine learning algorithms to enable the automated classification and precise identification of potentially cancerous areas. Additionally, the reliance on radiologist-identified cases introduces subjectivity, which could affect the generalizability of our results. Including ground-truth annotations obtained through biopsy or histopathology in future studies would provide a more objective basis for evaluation. Furthermore, exploring alternative segmentation techniques, such as texture analysis using co-occurrence matrices, may enhance segmentation accuracy. Lastly, our study employed a dataset with a limited number of patients, potentially affecting the representation of diverse breast cancer cases. A larger and more diverse dataset would provide a comprehensive evaluation of the proposed system’s performance and generalizability. Despite these limitations, our research demonstrates the potential of feature-based asymmetry analysis and its relevance in contributing to early breast cancer detection. Continued research and collaboration between medical professionals and technological experts holds promise for advancing diagnostic capabilities, ultimately improving patient outcomes in the fight against breast cancer. 

In conclusion, our research contributes to the advancement of early breast cancer detection through feature extraction and validation. The study underscores the significance of comparing contra-lateral breasts for asymmetry analysis, focusing on structural shape and skin thickness. The findings emphasize the potential clinical applications of our proposed system and the need for further exploration in this field. Advancements in technology and machine learning offer the opportunity to enhance structural analysis for early detection and monitoring. While our study primarily focused on feature extraction and validation, future research should explore the integration of machine learning algorithms for automated diagnosis and improved accuracy.

## Figures and Tables

**Figure 1 diagnostics-13-03440-f001:**
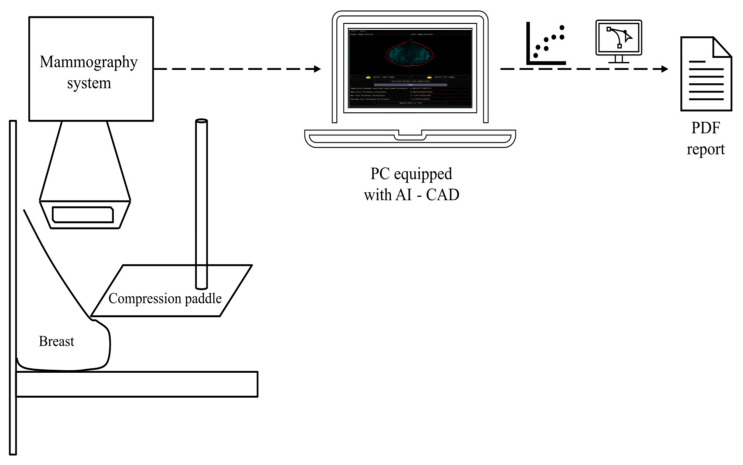
Schematic representation illustrating the integration of a CAD system with mammography equipment. The idea is that mammograms are extracted and submitted directly to the computer so that it can find patterns of asymmetry and interpret any abnormalities.

**Figure 2 diagnostics-13-03440-f002:**
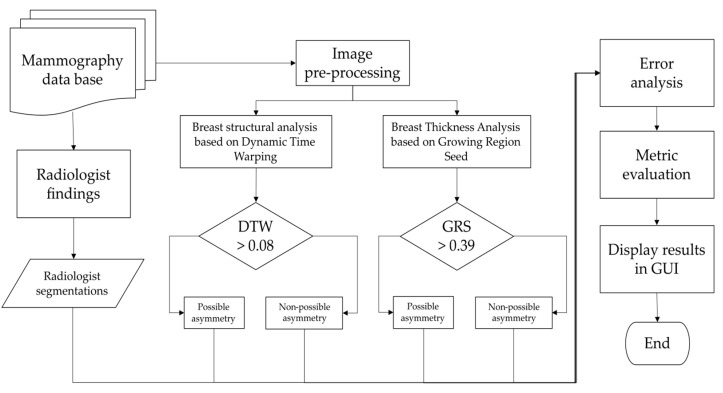
We propose hierarchical steps for breast image extraction from imaging to CAD system design. The process involves interpretation by specialists, image and data processing, evaluation of metrics, and presentation in a graphical interface.

**Figure 3 diagnostics-13-03440-f003:**
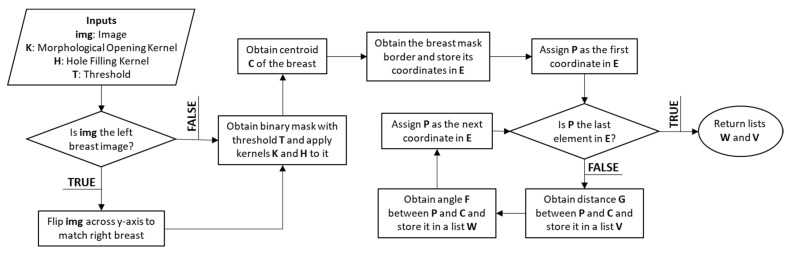
Flowchart illustrating the process of computing distances from the centroid to the breast’s edge.

**Figure 4 diagnostics-13-03440-f004:**
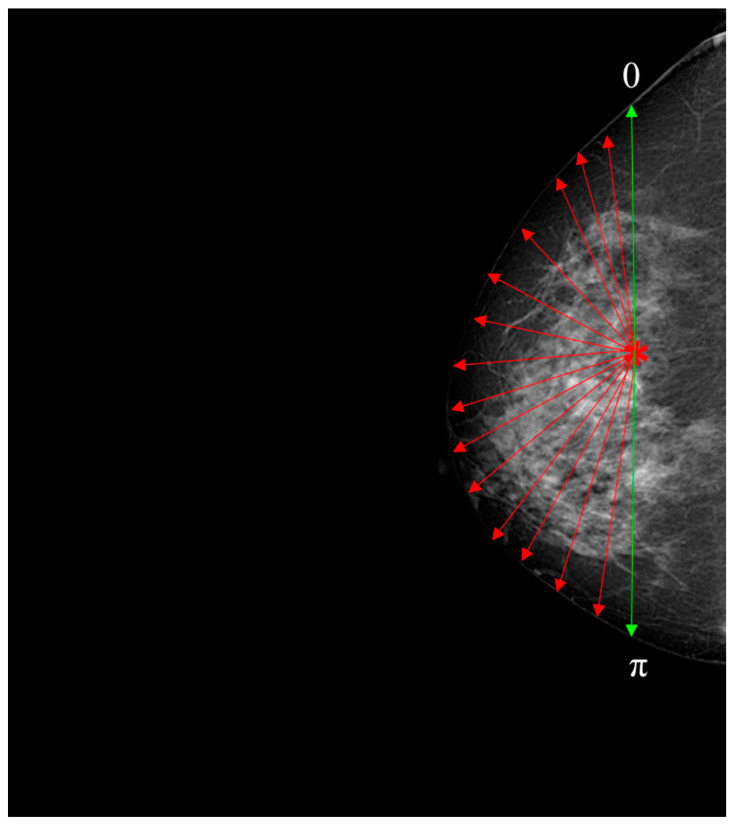
This diagram serves as an illustrative representation demonstrating the process of deriving a set of distances between the centroid and the periphery of the breast. The centroid is indicated by the asterisk symbol, while the red radius is depicted as arrows extending towards the edge of the breast and the green arrows indicate the beginning and ending [0–π].

**Figure 5 diagnostics-13-03440-f005:**
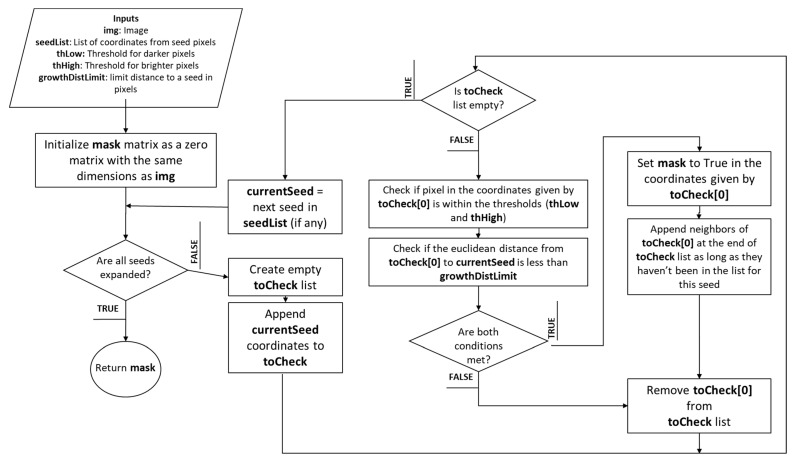
Flowchart representing the pseudocode for region growth through seed expansion.

**Figure 6 diagnostics-13-03440-f006:**
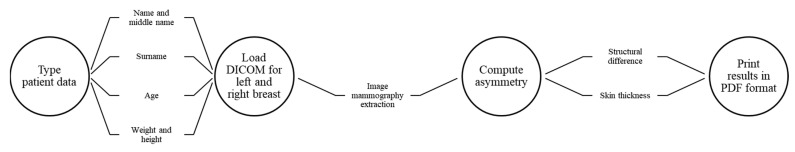
Flowchart depicting the GUI design for a CAD system used in medical diagnosis. The flowchart illustrates the sequential steps involved in the graphical user interface, including input fields for patient name, surname, age, weight, height, and the option to load mammography images for the left and right breast. The system generates a PDF report that includes the patient data, the loaded images, and the diagnosis result for possible breast asymmetry.

**Figure 7 diagnostics-13-03440-f007:**
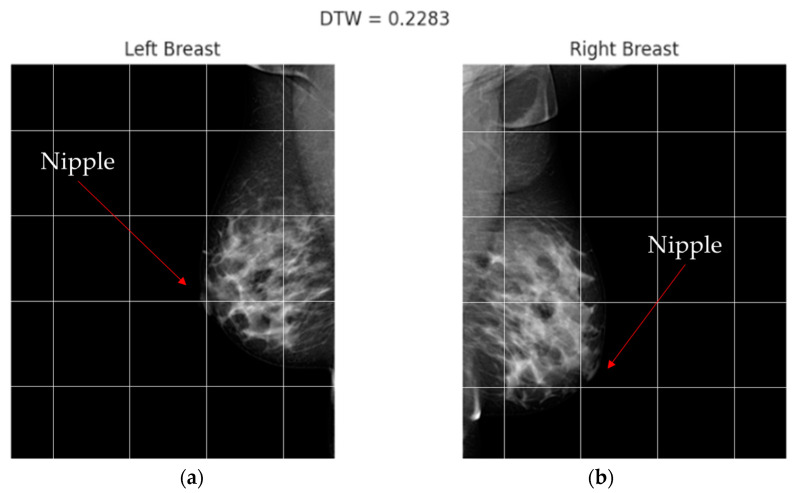
Shape asymmetry can be visible in the sample. Also, the disparity of the nipple is observable; (**a**) left CC projection and (**b**) right CC projection. The red arrow highlights the nipple region. By comparing the different regions relative to the grid, noticeable differences in shape and position can be observed. Additionally, using the pectoral muscles as a reference, it becomes evident that the right breast is positioned lower than the left one, further accentuating the asymmetry between the two breasts.

**Figure 8 diagnostics-13-03440-f008:**
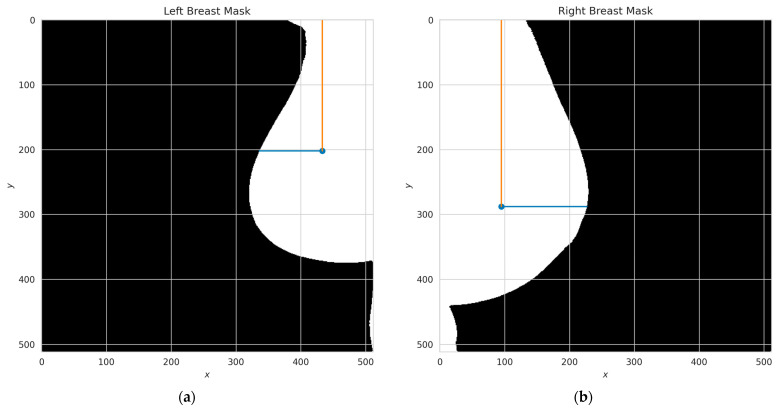
A plot of centroid–perimeter distances between the left and right breasts, in which it is possible to retrieve metrics such as Euclidian distance to compare as a first approach. If both breasts were similar, the distances should be similar; (**a**) centroid in left breast masking, (**b**) centroid in right breast masking.

**Figure 9 diagnostics-13-03440-f009:**
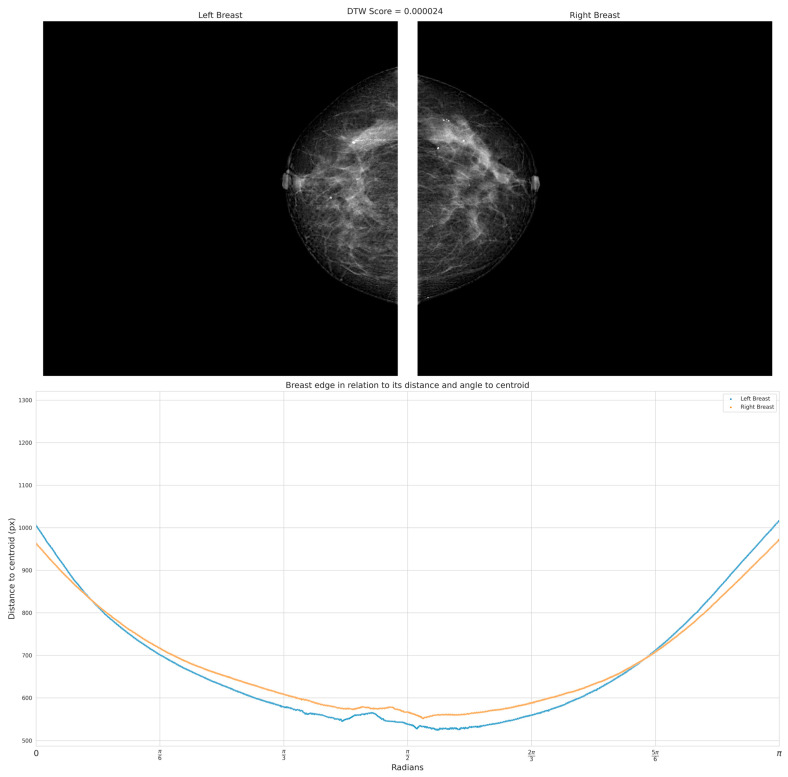
The results demonstrate a high degree of structural similarity between the breasts, indicating a lack of significant asymmetry. The curves generated from the analysis visually depict the close resemblance and alignment of the structural features. This finding supports the notion that no pronounced differences or variations exist between the corresponding regions of the breasts.

**Figure 10 diagnostics-13-03440-f010:**
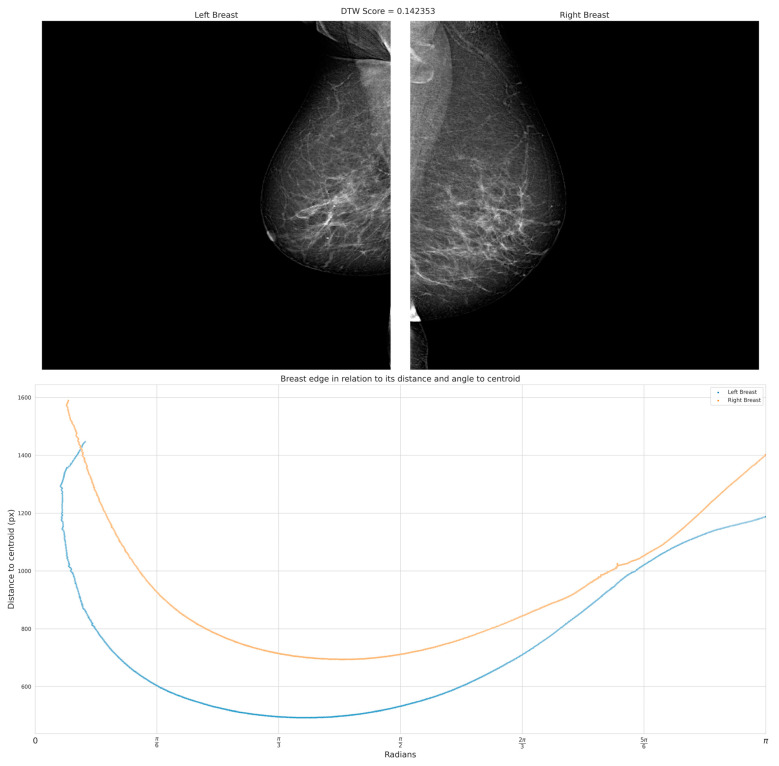
The analysis of this case reveals the presence of significant asymmetry, as indicated by a DTW index of 0.22. The generated result demonstrates notable differences and variations between the corresponding regions of the breasts. The observed asymmetry is visually apparent in the distinct patterns and shapes depicted in the curves.

**Figure 11 diagnostics-13-03440-f011:**
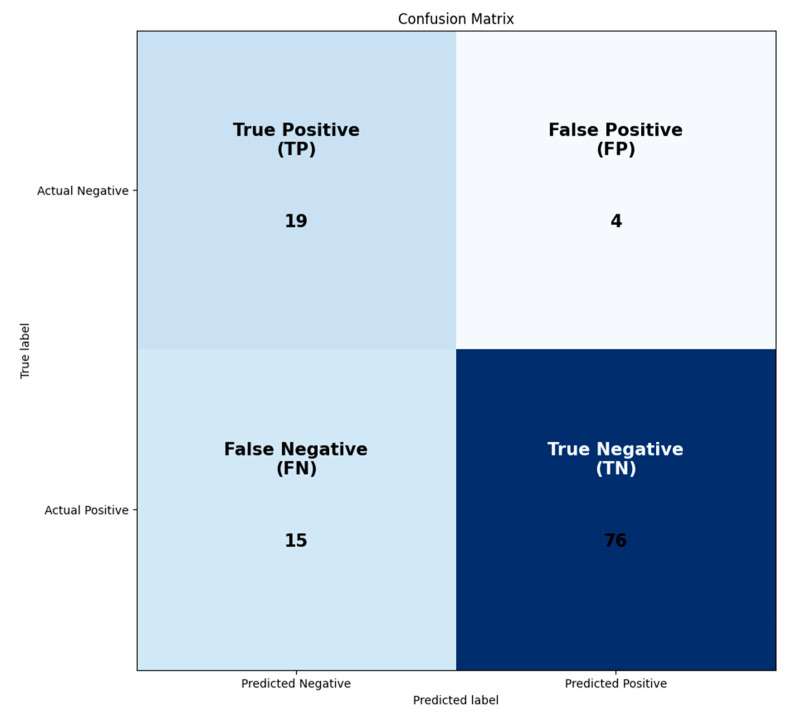
Confusion matrix for the DTW-based breast structure analysis algorithm.

**Figure 12 diagnostics-13-03440-f012:**
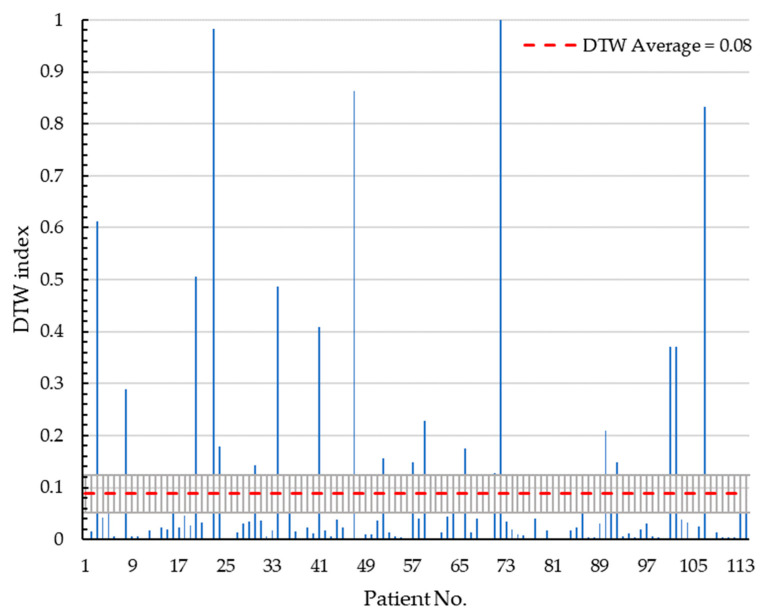
The DTW index results for each patient in the database are displayed in the graph. The graph includes a red segmented line representing the average value of µ = 0.08, with a standard deviation of std = 0.19. The DTW index values for each patient can be observed concerning the average line, providing insights into the level of dissimilarity or asymmetry present in the data.

**Figure 13 diagnostics-13-03440-f013:**
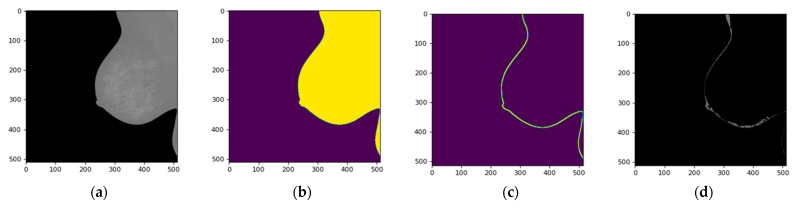
Skin segmentation results using GRS technique, (**a**) original projective CC-R image of a patient diagnosed with BI-RADS 1, (**b**) RoI detection mask, (**c**) seed placement based on threshold change detection, (**d**) seed growth based on average intensities of neighboring locations.

**Figure 14 diagnostics-13-03440-f014:**
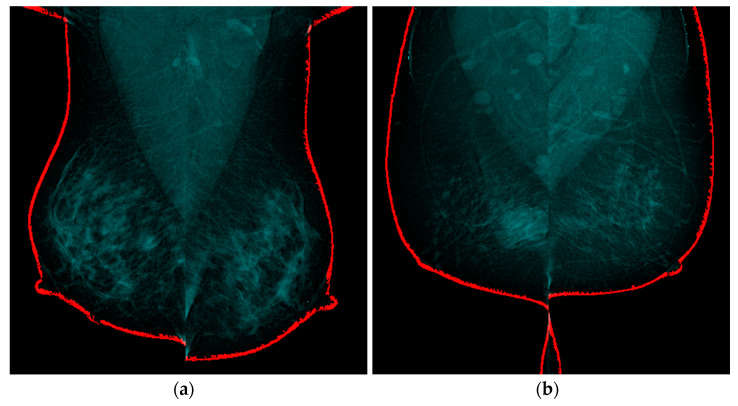
The results of the skin segmentation are depicted by the red-colored region, where (**a**) shows a patient with BIRADS 3, exhibiting an average difference of 0.004, indicating the best level of symmetry. On the other hand, (**b**) presents a patient with BIRADS 5, displaying an average difference of 1.199, suggesting the presence of asymmetry due to skin thickness. These findings highlight the ability of the segmentation algorithm to accurately identify and quantify variations in skin thickness, enabling the characterization of asymmetry in breast tissue.

**Figure 15 diagnostics-13-03440-f015:**
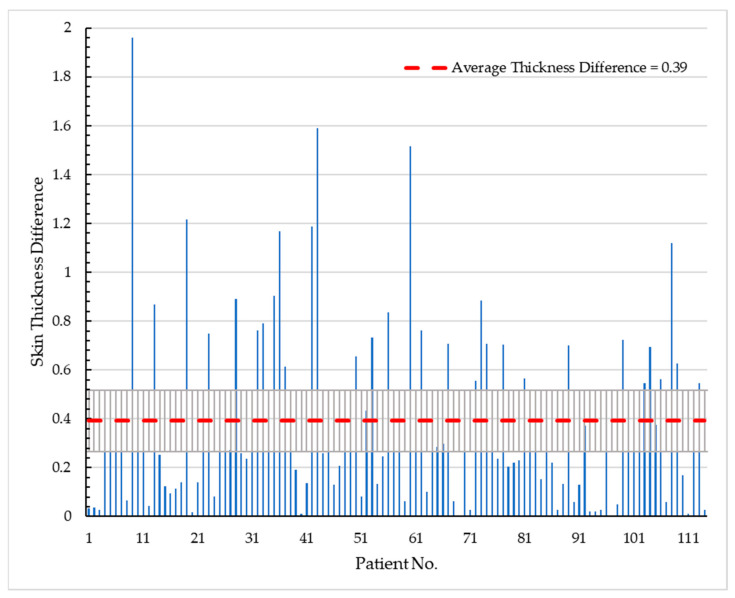
The results of the level of difference in skin thickness between contralateral sides for each patient are presented. An average threshold of µ = 0.39 was identified, with a standard deviation of std = 0.35. These findings indicate the extent of variation in skin thickness between the corresponding sides of the breasts. The threshold value provides a reference point for determining the presence of significant differences in skin thickness, aiding in the detection of potential asymmetries or abnormalities.

**Figure 16 diagnostics-13-03440-f016:**
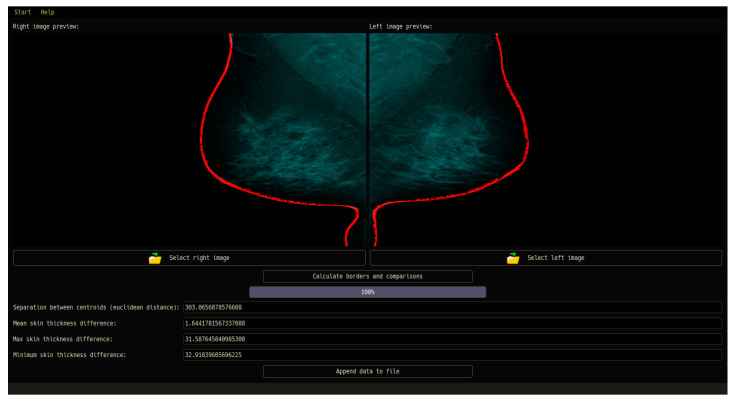
The GUI of the system enables the visualization of potential asymmetry cases. Users can upload mammograms in DICOM format, and the GUI seamlessly integrates with preprocessing algorithms, structural asymmetry analysis, and skin thickness difference computation. The specific case being referred to in this context involves a patient labeled as BIRADS 2, indicating a low suspicion of malignancy, yet exhibiting signs of possible asymmetry.

**Table 1 diagnostics-13-03440-t001:** The following table provides a summary of some of the most recent studies in which morphological asymmetries are proven to be signs of breast cancer.

Description	Characteristics	References
Results indicate that breast asymmetry is more likely to be a predictor of breast cancer rather than a result or consequence of the disease.	Breast volume and size.	Scutt D. Lancaster G. A., Manning J. T, 2005 [12,16]
The system employs medical imaging software to extract morphological features, involving the three-dimensional shape and size of both the breast and tumor.	Three-dimensional morphological features.	M. Singh, T. Singh, and S. Soni, 2021 [17]
Radiologists evaluated the images to assess the morphological features and enhancement characteristics.	Breast density.	Dawoud et al., 2022 [18]
Radiologists conducted a mammography descriptors evaluation such as shape and border characteristics, architectural distortion, and the presence of a hyperechoic rim and cystic complex structure.	Mass, chape, and microcalcifications.	Ghunaim et al., 2022 [19]
A deep learning model was proposed for bilateral asymmetrical detection based on Xception with two-dimensional tensors.	Bilateral asymmetrical detection.	Shimokawa et al., 2023 [20]

**Table 2 diagnostics-13-03440-t002:** Patient distribution is based on BI-RADS labeling.

BI-RADS Category	No. of Patients	Age Means	Standard Deviation
B1	14	50	10
B2	5	57	11
B3	70	58	9
B4	19	56	9
B5	6	55	5

**Table 3 diagnostics-13-03440-t003:** Cases found for asymmetry with respect to shape analysis with DTW.

BI-RADS	Number of Cases	Number of Asymmetries
1	24	6
2	10	2
3	34	7
4	34	6
5	11	1
6	1	1

**Table 4 diagnostics-13-03440-t004:** Parameters used to define the most optimal option.

Variable	Parameters
Upper intensity threshold	50, 100, 150, 255
Lower intensity threshold	1, 4, 7
Size *n* of the erosion kernel *n × n* for obtaining seeds	3, 6, 9, 12
Growing distance limit (maximum seed separation of seed from the edge)	3, 5, 7, 9

**Table 5 diagnostics-13-03440-t005:** Results of the parametric study in which the kernel size was varied for the erection and maximum growth distance. The best result is presented in bold—a 9 × 9 kernel 3 pixels deep.

Kernel for Erosion	Maximum Distance to Seed	RVD	ASSD
3.00	3.00	−0.8763	9.2908
3.00	5.00	−0.8441	9.1213
3.00	7.00	−0.8299	9.0694
3.00	9.00	−0.8236	9.0689
6.00	3.00	−0.3492	2.0468
6.00	5.00	−0.3014	2.0238
6.00	7.00	−0.2771	2.0334
6.00	9.00	−0.2651	2.0492
**9.00**	**3.00**	**0.1589**	**1.9182**
9.00	5.00	0.2380	1.9568
9.00	7.00	0.2848	2.0087
9.00	9.00	0.3096	2.0514
12.00	3.00	0.6451	2.0817
12.00	5.00	0.7327	2.1486
12.00	7.00	0.7857	2.2193
12.00	9.00	0.8129	2.2712

**Table 6 diagnostics-13-03440-t006:** The accuracy results are presented in terms of the average similarity to manually segmented images, along with the corresponding standard deviation.

Metric	Similarity Mean	Std. Deviation	Acceptable Similarity Cases (within Std. Deviation)	Non-Acceptable Similarity Cases (beyond Std. Deviation)
ASSD	1.887	2.313	171 (90.47%)	18 (9.52%)
RVD	0.158	0.344	126 (66.66%)	63 (33.33%)

## Data Availability

All data used in this research are available upon request from the corresponding author.

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
