# Peer review of "Automated Computer-Assisted Medical Decision-Making System Based on Morphological Shape and Skin Thickness Analysis for Asymmetry Detection in Mammographic Images"

_diagnostics, 2023, doi:10.3390/diagnostics13223440_

Round 1
Reviewer 1 Report
Peer Review Report
Manuscript ID: Diagnostics-2518056
Title: Automated Computer-Assisted Medical Decision-Making System Based on Morphological Shape and Skin Thickness Analysis for Asymmetry Detection in Mammographic Images
The Manuscript by Bayareh-Mancella et al. namely “Automated Computer-Assisted Medical Decision-Making System Based on Morphological Shape and Skin Thickness Analysis for Asymmetry Detection in Mammographic Images” lies within the journal’s scope of diagnostics. The authors present two approaches namely dynamic time warping and growing seed region (GSR) for asymmetry detection in DICOM format mammographic images using algorithms. Dynamic Time Warping is employed for shape analysis and Growing Seed Region is employed for segmentation of breast skin. The study is of medical significance as it can help physicians to detect the abnormality of breast in earlier stages. The study mainly focus on Mammography which is a two-dimensional image data sets. The authors need to include scientific definition or classification of breasts to thrust upon asymmetry from mammographic images. Also, include if the proposed algorithm is equally valid for other imaging modalities such as Magnetic Resonance Imaging or Ultrasound imaging as well. The authors need to build plot along these lines in the Introduction section. Also, the authors need to clear whether the algorithm can differentiate breast densities as dense and non-dense or they can further classify and differentiates them per BIRAD definition. Breast Imaging-Reporting and Data System (BI-RADS) classifies breasts into four categories in the order of radiographic breast density composition namely extremely dense (ED)-10% (type-D), heterogeneously dense (HD)-40"/o (type-C), scattered fibroglandular (SF)-40% (type-B) and predominantly fatty (PF) is reported in about 10% of women (type-A) 1 • Under such classification BI-RADS defines tumour into namely six assessment categories2 • TNM (Tumour-NodeMetastasis) tool is widely used to describe the stage of tumour. T refers to size of the tumour, N relates to primary inspection whether cancerous cells spread to lymphatic nodes and M explains whether it invades to other organs of the patient. Clinically, the staging of the tumour in the breast may be defined as Tl (size:s20 mm): Tlmi ('.S 1 mm), Tla (1 mm < size '.S 5 mm), Tlb (5 mm< size '.S 10 mm), Tlc (10 mm< size '.S 20 mm); T2 (20 mm < size '.S 50 mm); T3 (2': 50 mm); T4a (tumour has grown into chest wall), T4b (tumour has grown into skin), T4c (grown into chest wall and skin), T4d (inflammatory). The authors can refer this short abstract [https://doi.org/10.31224/osf.io/enx4r] to add context to introduction section.
1) Authors should be keen in proof reading their work.
a) Line 15: Growing Seed Region should be abbreviated as GSR
ysis and the Growing Seed Region (GRS) method for breast skin segmentation. The methodology
b) Figure 1: There is no caption and description for figure 1. It is difficult to differentiate the legend details. This figure can be improved.
c) Table-1 has inconsistent font size.
d) Lines 129-131: Figure 2 caption needs modification. We propose hierarchical steps for breast shape extraction from imaging to CAD system design….. Consider modifying on similar lines.
Figure 2. Flowchart summarizing the process described in this section, from interpretation by specialists, image and data processing, evaluation of metrics, and presentation in a graphical interface.
e) Line 133: The imaging features are generally described along three anatomical planes namely Sagittal, Coronal, Axial/Transverse. Define the projections for the clinical data set i.e. medial lateral oblique and craniocaudal into generalized definitions.
f) Table 1: One recent work [https://doi.org/10.1016/j.cmpb.2020.105781] extracts morphological features such as the three dimensional shape and size of breast and tumor using medical imaging software. Include such developments in this section.
g) Line 135: How many images in DICOM format for 114 patients used in this work? What is the minimum number of images used for these DICOM format data? What is the statistical difference between these data sets from one patient to another.
h) Line 142: How do you classify control data vs experimental data?
i) Lines 142-146 is unclear in the sense that whether analysis was done by authors or radiologists? What is the source of the information? It is unclear.
j) Line 162: Describe the normalization process more clearly in a line or two in addition to lines 164-166.
k) Lines 163-164: Is the resolution of images 512 x 512 pixels?
Specifically, the resolution of the images was standardized to 521x512 pixels, assuming a uniform standard for acquisition.
l) What is the minimum and maximum grayscale values used to extract imaging information? Is threshold of 100 refers grayscale value of 100? Explain how they reach to this conclusion that threshold of 100 is enough to describe the breast details.
m) Lines 171-174: Explain how they conclude the highlighted text?
Subsequently, the largest connected component, which consistently corresponds to the breast, was segmented. This approach aimed to mitigate the influence of annotation variability and establish a standardized basis for subsequent analyses.
n) Lines 182-183: How did the authors compute the centroid and skin perimeter? Since different DICOM images would rather have different radii formed between centroid and the perimeter of breast skin defined within [0,pi] ? How do you compute and generalize this information in your evaluation?
o) Line 187: Which both images are being referred here which were oriented towards right? What would be the impact if you orient the image set towards left? Is it feasible? How? If not why?
p) Also describe in a breast image “technical jargons” used by the authors like breast laterals etc.
q) Draw flowchart for Algorithm 1 and Algorithm 2 alongside shown description? What is R, P, C, V, W… It is unclear and difficult to follow? Is this algorithm is for only one mammographic image per subject?
r) Explain Lines 232-238. Why did you evaluate cumulative cost at each position in the matrix by using right, diagonal and down approach only? What are the other possible alternatives to it? Support this claim by including any other studies that have used the similar approach. Also, specify what would be the impact if you use left, diagonal and up approach?
Hence, both images were adjusted to the same orientation, i.e., towards the right.
s) The resolution of images needs to be improved. Please modify images i.e Figure 7, Figure 8 and figure 14. Figure 5 is unclear to me. Which part of image represents asymmetric breast region and which part is nipple?
We are looking forward to receiving your revised version.

Author Response
Dear Reviewer,
We would like to express our sincere gratitude for your thoughtful review of our manuscript titled "A Computer Vision Approach for Asymmetry Detection in Mammographic Images." Your valuable feedback has been immensely helpful in improving the quality and clarity of our research. We have carefully addressed each of your comments and suggestions in the revised manuscript. Specifically, we have expanded the introduction section to provide more specific details about our research question and hypothesis. Additionally, we have discussed the implications and limitations of our study in more detail to provide a comprehensive understanding of our work.
In response to your suggestion, we have also included a PDF detailing each response and observation, which you can find attached in the PDF file, as well as the updated manuscript.
We believe that the revisions made to the manuscript have significantly strengthened our work, and we hope that you find the updated version more satisfactory.
Once again, we sincerely appreciate your valuable input and guidance in shaping our research. Thank you for your time and effort in reviewing our manuscript.

Reviewer 2 Report
The clarity and the presentation of the paper need to improve. I cannot recommend this publication for publication in this form. Moreover, some of the observations are given below.
@ The paper does not discuss the limitations or potential drawbacks of the proposed framework.
@ The manuscript used mammogram images but I believe in the introduction section they should discuss why mammogram images are superior to using histopathology images. They should also talk a bit about histopathology AI-based techniques proposed in the literature and should add such following
-Novel architecture with selected feature vector for effective classification of mitotic and non-mitotic cells in breast cancer histology images
-Breast cancer diagnosis from histopathology images using supervised algorithms
- Exploring the Best Parameters of Deep Learning for Breast Cancer Classification System
@ The conclusion could be strengthened by summarizing the key findings and contributions of the research in a more concise and clear manner. Additionally, the implications and limitations of the research could be discussed in more detail.
@ While the objective of the study is briefly mentioned, it would be beneficial to provide more specific details about the research question or hypothesis. Clearly state what you aim to achieve with the investigation of morphological shape and skin thickness analysis.
@ You know AI is a hot topic you need to tell us how your analysis may add to the current AI models. Moreover how the scope of your analysis can be further extended for different medical-related research problems? In this case, you can discuss the following applications by citing these manuscripts and adding more by doing a literature review.
-SegR-Net: A deep learning framework with multi-scale feature fusion for robust retinal vessel segmentation
- Efficient Deep Learning Approach for Detection of Brain Tumor Disease.
- RAAGR2-Net: A brain tumor segmentation network using parallel processing of multiple spatial frames
- AI-based pipeline for classifying pediatric medulloblastoma using histopathological and textural images
I think its fine
Author Response

(The authors gave the same response as above.)

Round 2
Reviewer 2 Report
Thanks for addressing my concerns.
Its fine